# Structural basis for histone variant H3tK27me3 recognition by PHF1 and PHF19

**Cheng Dong[1], Reiko Nakagawa[2], Kyohei Oyama[3], Yusuke Yamamoto[3], Weilian Zhang[4], Aiping Dong[4], Yanjun Li[4], Yuriko Yoshimura[5], Hiroyuki Kamiya[3], Jun-ichi Nakayama[5,6], Jun Ueda[7], Jinrong Min[4,8]\***

[1]Department of Biochemistry and Molecular Biology, School of Basic Medical Sciences, Tianjin Medical University, Tianjin, China; [2]Laboratory for Phyloinformatics, RIKEN Center for Biosystems Dynamics Research, Kobe, Japan; [3]Department of Cardiac Surgery, Asahikawa Medical University, Asahikawa, Japan; [4]Structural Genomics Consortium, University of Toronto, Toronto, Canada; [5]Division of Chromatin Regulation, National Institute for Basic Biology, Okazaki, Japan; [6]Department of Basic Biology, School of Life Science, The Graduate University for Advanced Studies (SOKENDAI), Okazaki, Japan; [7]Centre for Advanced Research and Education, Asahikawa Medical University, Asahikawa, Japan; [8]Department of Physiology, University of Toronto, Toronto, Canada

**Abstract** The Polycomb repressive complex 2 (PRC2) is a multicomponent histone H3K27 methyltransferase complex, best known for silencing the *Hox* genes during embryonic development. The Polycomb-like proteins PHF1, MTF2, and PHF19 are critical components of PRC2 by stimulating its catalytic activity in embryonic stem cells. The Tudor domains of PHF1/19 have been previously shown to be readers of H3K36me3 in vitro. However, some other studies suggest that PHF1 and PHF19 co-localize with the H3K27me3 mark but not H3K36me3 in cells. Here, we provide further evidence that PHF1 co-localizes with H3t in testis and its Tudor domain preferentially binds to H3tK27me3 over canonical H3K27me3 in vitro. Our complex structures of the Tudor domains of PHF1 and PHF19 with H3tK27me3 shed light on the molecular basis for preferential recognition of H3tK27me3 by PHF1 and PHF19 over canonical H3K27me3, implicating that H3tK27me3 might be a physiological ligand of PHF1/19.

**\*For correspondence:**
jr.min@utoronto.ca

**Competing interests:** The authors declare that no competing interests exist.

## Introduction

Polycomb group (PcG) proteins act as transcriptional repressors and are involved in multiple cellular processes including cell fate determination, cell cycle, X chromosome inactivation, genomic imprinting, and tumorigenesis (*Bracken and Helin, 2009*; *Gieni and Hendzel, 2009*; *Sparmann and van Lohuizen, 2006*; *Surface et al., 2010*). Two main PcG protein complexes have been characterized, known as PRC1 and PRC2 (Polycomb repressive complex 1 and 2). PRC2 is responsible for trimethylation of K27 on histone H3 (*Cao et al., 2002*; *Czermin et al., 2002*; *Müller et al., 2002*), generating a silencing chromatin mark H3K27me3 that can be read out by Pc, a component of the PRC1 complex (*Cao et al., 2002*; *Fischle et al., 2003*; *Min et al., 2003*). In addition to the core components (EZH1/2, EED, SUZ12, and RBBP4/7), the PRC2 complex also contains some accessory components, such as JARID and PCL (Polycomb-like), which contribute to the enzymatic activity of PRC2 as well (*Margueron and Reinberg, 2011*; *Sauvageau and Sauvageau, 2010*; *Schuettengruber et al., 2017*).

The *PCL* gene was initially identified in Drosophila (*Duncan, 1982*). In humans, there are three PCL homologs, namely, PCL1, PCL2, and PCL3 (also known as PHF1, MTF2, and PHF19, respectively). In vivo and in vitro studies reveal that PHF1 stimulates the catalytic activity of PRC2 and plays an important role in the silencing of the *Hox* genes (*Cao et al., 2008*; *Nekrasov et al., 2007*; *Sarma et al., 2008*). PHF1 is also involved in response to DNA double-strand breaks (DSBs) through its interaction with Ku70/Ku80 in human cells (*Hong et al., 2008*) and regulates cellular quiescence and apoptosis by interacting with P53 (*Brien et al., 2015*; *Yang et al., 2013*). MTF2 is expressed abundantly in embryonic stem (ES) cells, wherein it regulates ES cell self-renewal, pluripotency, and cell fate decision (*Walker et al., 2010*; *Walker et al., 2011*). MTF2 also modulates somatic cell reprogramming as a key component of the PRC2 complex (*Zhang et al., 2011*). PHF19 modulates the PRC2 function by promoting H3K27 trimethylation, contributing to ES cell self-renewal (*Hunkapiller et al., 2012*). Overexpression of PHF1 and PHF19 is associated with many types of cancers (*Liu et al., 2018*; *Wang et al., 2004*).

The three human PCL proteins all contain a single Tudor domain followed by two plant homeodomain (PHD) fingers and an extended homologous (EH) domain (*Figure 1A*). The Tudor domains of PHF1 and PHF19 have been shown to be 'readers' of H3K36me3 in vitro and link PRC2 to the chromatin (*Ballaré et al., 2012*; *Brien et al., 2012*; *Cai et al., 2013*; *Musselman et al., 2012*; *Qin et al., 2013*). The PHD1 domain of PHF1 can recognize symmetric dimethylation of H4R3 (H4R3me2s; *Liu et al., 2018*). The EH domain, which binds to DNA, mediates the recruitment of PRC2 to CpG islands (*Li et al., 2017*). Recently, the C-terminal domain of PHF19 stabilizes the dimerization of intrinsic PRC2 by interacting with SUZ12 and enhances the PRC2 binding to CpG islands (*Chen et al., 2020*; *Chen et al., 2018*). Previous data have suggested that PHF1 stimulates the H3K27me3 catalytic activity of PRC2 (*Cao et al., 2008*; *Sarma et al., 2008*) and H3K36me3 recognition by PHF1 promotes the spreading of PRC2 and H3K27me3 into H3K36me3-marked loci (*Cai et al., 2013*). Similarly, the PHF19-H3K36me3 interaction is required for the full enzymatic activity of PRC2 and responsible for the recruitment of PRC2 and H3K36me3 demethylases NO66 and/or KDM2b to the target loci to remove the H3K36me3 active mark and facilitate deposition of H3K27me3 (*Abed and Jones, 2012*; *Ballaré et al., 2012*; *Brien et al., 2012*). On the other hand, a study has shown that PHF19 does not co-localize with H3K36me3 but is associated with H3K27me3 in mouse ES cells and binds to H3K27me3 in vitro, albeit weaker than to H3K36me3 (*Brien et al., 2012*). Likewise, PHF1 does not co-localize with H3K36me3 in cells and recognizes the testis-specific histone H3tK27me3 mark as well as H3K36me3 in vitro (*Kycia et al., 2014*). In fact, the inhibition of PRC2 activity by H3K36me3 is well documented (*Schmitges et al., 2011*; *Voigt et al., 2012*; *Yuan et al., 2011*) and the in vitro enzymatic assays also show that binding of PHF1 to H3K36me3 inhibits methyltransferase activity of the PRC2 complex (*Musselman et al., 2012*). To reconcile the inconsistency among these studies, we systematically investigated the binding selectivity of the PCL family of proteins to H3K36me3, H3K27me3, and H3tK27me3, and found that the Tudor domains of PHF1 and PHF19 bound to H3tK27me3 were slightly weaker than to H3K36me3 but they preferentially recognized H3tK27me3 over canonical H3K27me3. We also showed that PHF1 was co-localized with H3t during the pachytene and diplotene stages as well as the round spermatid stage. Our crystal structures of the Tudor domains of PHF1 and PHF19, respectively, in complex with H3tK27me3 shed light on the molecular basis for preferential recognition of H3tK27me3 by PHF1 and PHF19 over canonical H3K27me3.

## Results

### Tudor domains of PHF1 and PHF19 preferentially bind to testis-specific H3tK27me3 over canonical H3K27me3 in vitro

In order to further explore the substrate specificity of the three human PCL Tudor domains, we employed isothermal titration calorimetry (ITC) to examine their binding affinities with three different peptides, that is, H3K36me3, H3K27me3, and H3tK27me3 (*Figure 1B*). In agreement with previous reports (*Ballaré et al., 2012*; *Brien et al., 2012*; *Kycia et al., 2014*; *Musselman et al., 2012*), the Tudor domains of PHF1 and PHF19 were bound to H3K36me3. In addition, they were also bound to H3tK27me3, albeit ~twofold weaker than to H3K36me3 but ~fivefold stronger than to canonical H3K27me3 (*Figure 1D and E*). The Tudor domain of MTF2 exhibited significantly weaker binding to

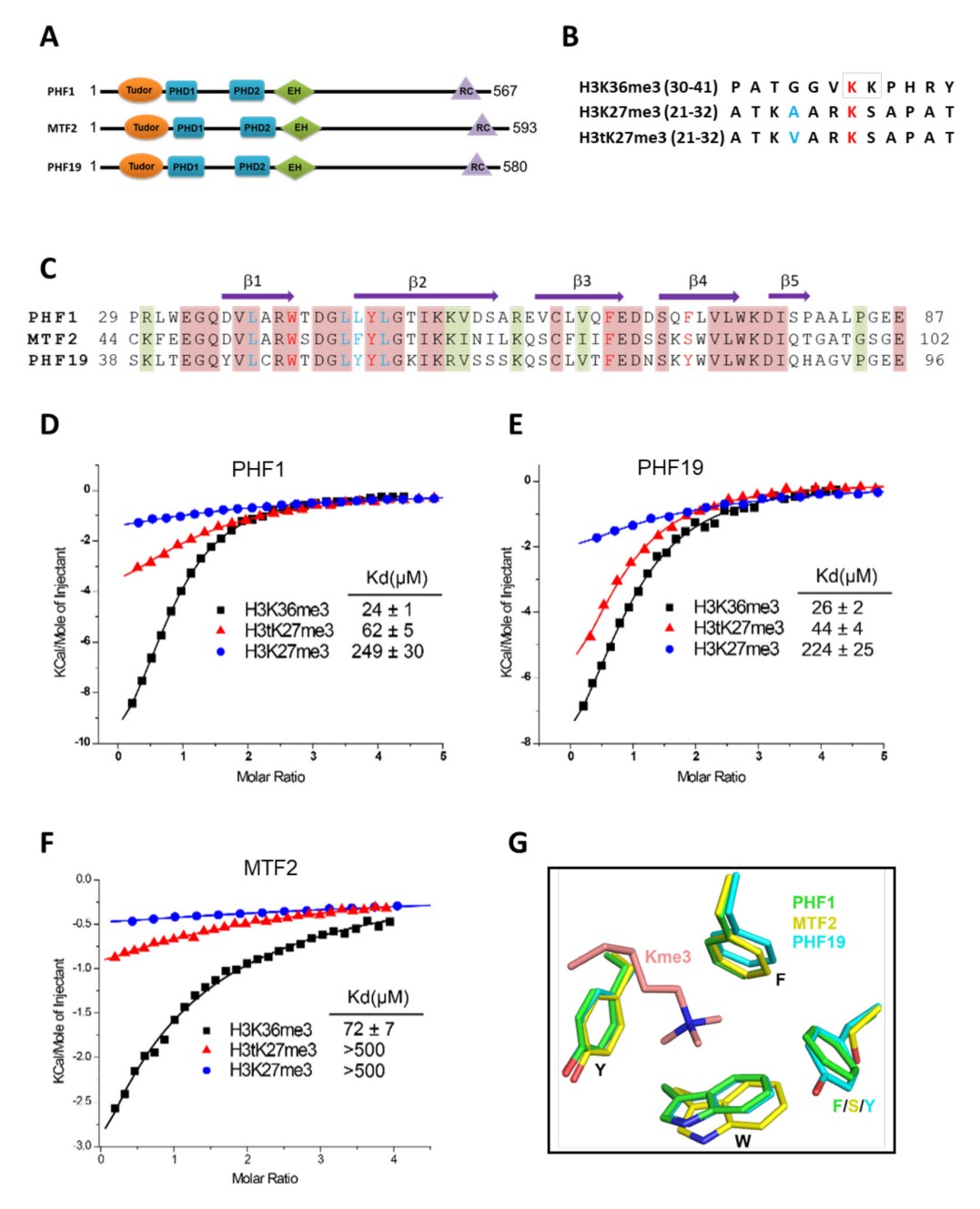

**Figure 1.** Tudor domains of PHF1 and PHF19 preferentially recognize H3K36me3 and H3tK27me3. (**A**) Schematic representation of domain structures of PHF1, MTF2, and PHF19. PHD: plant homeodomain finger. EH: extended homologous domain. (**B**) Peptide sequences of H3K36me3, H3K27me3, and H3tK27me3. The methylated lysines are colored in red. A24 of histone H3 and V24 of histone H3t are colored in blue. (**C**) Sequence alignment of the Tudor domains of PHF1, MTF2, and PHF19. The secondary structure elements are shown above the alignment. Identical residues and conserved residues are highlighted in red and olive-green backgrounds, respectively. The residues forming the aromatic cage and the leucyl patch are colored in red and blue, respectively. (**D-F**) ITC measurements of the interactions of the Tudor domains of PHF1, PHF19, and MTF2 with the H3K36me3, H3tK27me3, and H3K27me3 peptides. (**G**) Superposition of the aromatic cage residues of PHF1-H3tK27me3 complex (green, this study), MTF2-H3K36me3 (yellow, PDB: 5XFR), and PHF19-H3tK27me3 complex (cyan, this study). The trimethyl-lysine is shown as salmon sticks.

these peptides than those of PHF1 and PHF19 (*Figure 1F*). Based on the sequence alignment of these three proteins (*Figure 1C*), we noticed that the aromatic cage of MTF2 is composed of only three aromatic residues, unlike PHF1 and PHF19, whose cages are formed by four aromatic residues. Ser86 in MTF2 takes the place of an aromatic residue, which is Phe71 in PHF1 or Tyr80 in PHF19,

respectively (*Figure 1G*). This incomplete aromatic cage might cause weak binding ability of H3tK27me3, similar to that of H3K36me3 (*Gatchalian et al., 2015*). Indeed, mutating Ser86 of MTF2 to phenylalanine or tyrosine resulted in dramatically enhanced binding affinities (*Table 1*). Thus, a complete aromatic cage is required for efficient histone binding. Taken together, our results confirm that the Tudor domains of PHF1 and PHF19 were bound to both H3K36me3 and H3tK27me3 and preferentially bound to testis-specific H3tK27me3 over canonical H3K27me3 in vitro.

## PHF1 and H3tK27 methylation are linked in vivo

To investigate the association between PHF1 and H3tK27me3 in cells, we first examined the expression patterns of PHF1 in various mouse tissues. Similar to histone H3t, PHF1 was highly expressed in testis (*Figure 2A*). In agreement with our observation, mouse PHF1 (also known as Tctex3) is also highly expressed in testis (*Kawakami et al., 1998*). Next, we examined the expression timing of PHF1 during spermatogenesis and found that PHF1 started to accumulate in the nucleus from the pachytene stage to the round spermatid stage during meiosis (*Figure 2C*). By immunostaining of the testis sections, we found that PHF1 co-localized with H3t during the pachytene and diplotene stages as well as the round spermatid stage (*Figure 2B and D*). Moreover, like H3t, PHF1 was excluded from the XY body (*Figure 2E*). To further explore their interactions, we performed GST pull-down assays using histones prepared from the H3t-overexpressed HEK293T cells. We utilized HEK293T cells to transiently express H3t since the percentage of cells expressing H3t in testis is low. As the fraction extracted using sulfuric acid contains H3t, we could assume that H3t transiently expressed in HEK293T cells has been incorporated into the chromatin. Our pull-down results showed that PHF1-Tudor, but not GST alone, pulled-down histones H3 and H3t (*Figure 3A–B*). Remarkably, LC-MS/MS analysis detected H3tK27-methylated fragments in the pull-down samples while such fragments were hardly detected in the input samples (*Figure 3C*). To evaluate the relative abundance of H3t containing either K27 or K36 methylation, we have tried several different peptidases, but we could not efficiently detect H3t fragments containing both V24 and methylated K36 presumably due to that fact that unmodified K27 would be cleaved during the peptide digestion processes. Therefore, we compared the modification status of K27 and K36 in input and pull-down samples regardless of the H3 species. We found that the H3K27me2/3 peptides were strongly enriched when pull-down with the GST tagged PHF1-Tudor protein. By contrast, the H3K36me3 peptide was not noticeably enriched in the pull-down samples (*Figure 3D* and *Supplementary file 1*). Therefore, we suggest that PHF1-Tudor can bind to methylated H3tK27 in vitro and in vivo.

**Table 1.** Binding affinities of PHF1/MTF2 mutants with the H3tK27me3 peptide.

| PHF1 | ITC (µM) |
| --- | --- |
| W41A | NB |
| Y47A | NB |
| F65A | NB |
| F71A | 192 ± 18 |
| L38A | 125 ± 9 |
| L45A | 178 ± 9 |
| L46A | 127 ± 8 |
| L46Y | 50 ± 5 |
| L48A | 81 ± 2 |
| L38A, L45A, L46A, L48A | >500 |
| E66A | >200 |
| MTF2 | ITC (µM) |
| S86F | 91 ± 21 |
| S86Y | 110 ± 11 |

NB indicates no detectable binding.

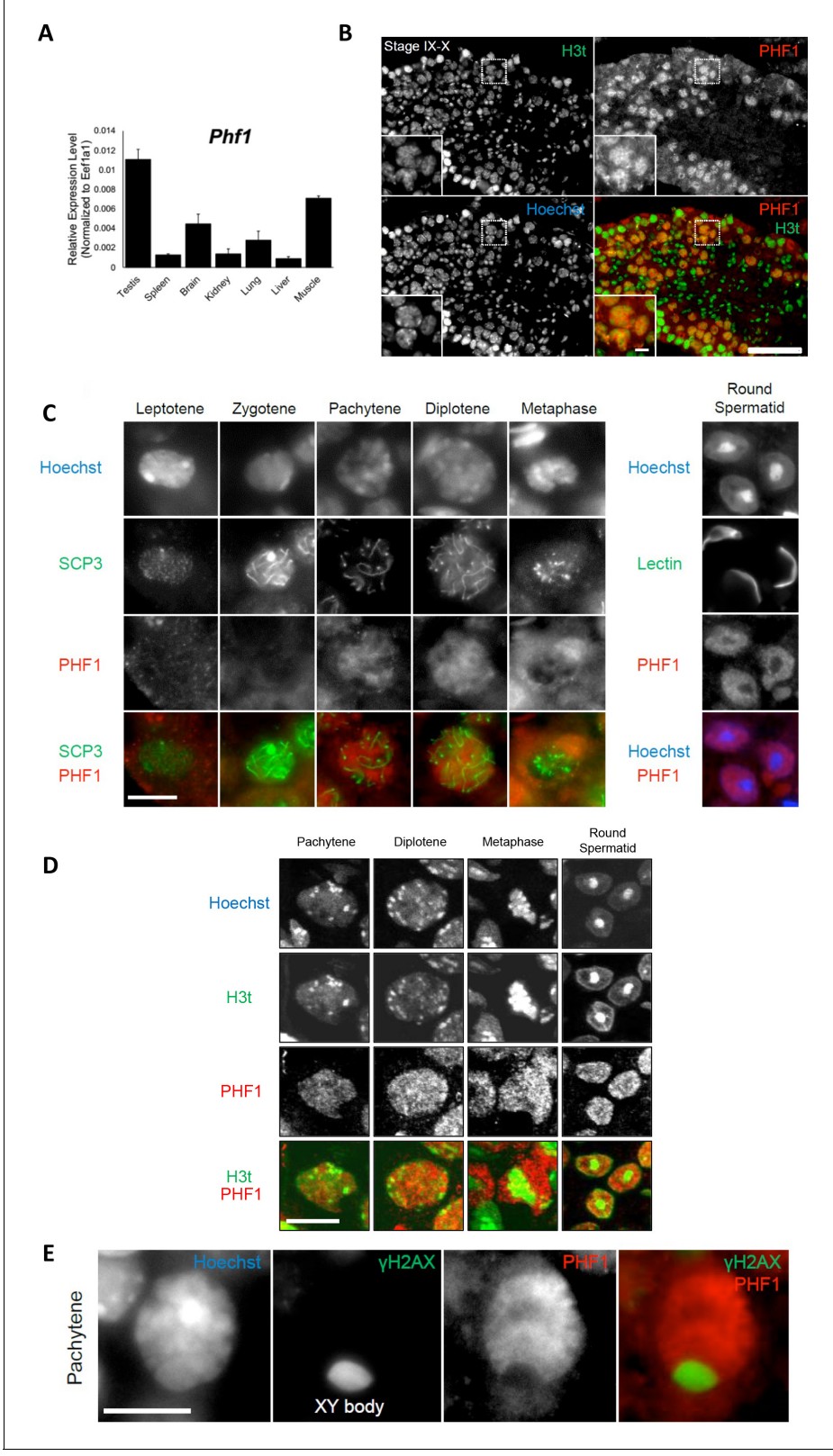

**Figure 2.** PHF1 is associated with the H3t in vivo. (**A**) Expression levels of PHF1 in different mouse tissues using the RT-qPCR analysis. (**B**) Immunostaining of testis sections by antibodies against H3t (green) and PHF1 (red). Seminiferous tubules at stages IX-X are indicated and magnified images of the pachytene stage nuclei are shown in each panel. Scale bars, 100 µm and 10 µm, in magnified image. (**C**) PHF1 expression during meiosis. Note that PHF1 started to accumulate in the nucleus from the pachytene stage. Scale bar, 10 µm. (**D**) PHF1 (red) and H3t (green) co-localize from the

*Figure 2 continued on next page*

Figure 2 continued

pachytene until the round spermatid stage. Scale bar, 10 μm. (E) PHF1 was excluded from the XY body. Testis sections were stained with anti-PHF1 (red) and anti–γH2AX (green) antibodies and the pachytene stage nucleus is indicated. H3t is also excluded from the XY body. Scale bar, 10 μm.

## Structural basis for preferential binding of PHF1/PHF19 to H3tK27me3

To elucidate the molecular basis for the recognition of H3tK27me3 by PHF1 and PHF19, we determined the crystal structures of PHF1-Tudor and PHF19-Tudor in complex with H3tK27me3, respectively (*Table 2*). In the complex structures, PHF1 and PHF19, adopt a classic Tudor domain fold with a five-stranded antiparallel β-barrel (*Figure 4A and B*). The binding mode of PHF1 or PHF19 to the H3tK27me3 peptide, especially the central VARKme3 motif, is highly conserved (*Figure 4C and D*). The conserved portion of the interaction surface comprises a hydrophobic leucyl patch, an aromatic cage, and a negatively charged groove (*Figure 4E and F*). Most of the interacting residues are conserved between PHF1 and PHF19 (*Figure 1C*). Taking the PHF1-H3tK27me3 complex as an example, A25 is stabilized by hydrophobic interactions with Leu45 and Tyr47. The amino group of A25 also forms a hydrogen bond with the Leu46 carbonyl oxygen, and the A25 carbonyl oxygen participates in a network of solvent-mediated hydrogen bonds with Leu48 and Glu66. The trimethylated K27 residue inserts into the aromatic cage, where the positively charged trimethyl ammonium group is surrounded by the aromatic sides of Trp41, Tyr47, Phe65, and Phe71. Asp67 and Ser69 also contribute to the K27me3 recognition by providing some polar interactions (*Figure 4C*). The importance of these interacting residues has been confirmed by our mutant binding data. Mutating Trp41, Tyr47, Phe65, or Glu66 of PHF1 to alanine disrupted or significantly decreased its binding to the H3tK27me3 peptide, whereas mutation of the more peripheral F71 displayed threefold weaker binding (*Table 1*).

The testis-specific H3tK27me3 and canonical H3K27me3 peptide sequences differ only in one position, that is, V24 of H3t versus A24 of H3 (*Figure 1B*). In the PHF1-H3tK27me3 complex, V24 is coordinated by a hydrophobic leucyl patch comprising Leu38, Leu45, Leu46, and Leu48, and Leu38 and Leu45 form an L-L clasp interacting with H3t-V24 (*Figure 4C*). Compared to A24 in H3, the side chain of H3tV24 enables more extensive interactions with the PHF1 leucyl patch. The single point mutants L38A, L45A, or L46A of PHF1 exhibited ~two to threefold reduced binding affinities to H3tK27me3, and mutation of the more peripheral L48 also displayed slightly weaker binding (*Table 1*). The leucyl patch is conserved in PHF19, except that Leu46 in PHF1 is replaced with Tyr55 in PHF19 (*Figure 1C*). Tyr55 of PHF19 might provide even stronger hydrophobic interactions than Leu46 of PHF1 with V24 (*Figure 4D*), which might explain why PHF19 has a stronger interaction with H3tK27me3 than PHF1 (*Figure 1D,E*). Indeed, the L46Y mutant of PHF1 displays an increased binding affinity to H3tK27me3 (*Table 1*). Therefore, PHF1 and PHF19 bind more tightly to H3tK27me3 than to canonical H3K27me3 and the hydrophobic leucyl patch is the structural determinant.

## H3tK27me3 and H3K36me3 peptides bound to PHF1/19 within the same binding grooves but in inverse directions

The complex structures of PHF1-Tudor and PHF19-Tudor, respectively, with a H3K36me3 peptide have been determined previously (*Ballaré et al., 2012*; *Cai et al., 2013*; *Li et al., 2017*; *Musselman et al., 2012*). Compared to our complex structures, we found that both H3tK27me3 and H3K36me3 peptides reside within the same binding grooves of PHF1 and PHF19, but their backbones are reversed (*Figure 5A and B*). Nevertheless, both peptides interact with a similar set of residues from PHF1/19 (*Figure 5C and D*). Moreover, the *N*-VARKme3-*C* fragment of H3tK27me3 could be overlaid with the *C*-HPKKme3-*N* fragment of H3K36me3 (*Figure 5E*). Specifically, V24 of H3tK27me3 and H39 of H3K36me3 share the same binding pocket and are bound by the hydrophobic leucyl patch; A25 of H3tK27me3 and P38 of H3K36me3 occupy the same site; R26 of H3tK27me3 and K37 of H3K36me3 are recognized by salt bridge interactions; and both K27me3 and K36me3 are buried in the same aromatic cage (*Figure 5C and D*). This binding direction promiscuity might also explain why PHF1/19 showed some weak binding to H3K4me3, H3K9me3, H3K27me3, and H4K20me3 peptides (*Musselman et al., 2012*), whose sequences diverge significantly from the sequence of H3K36me3 but align better in reverse directions (*Figure 5E*).

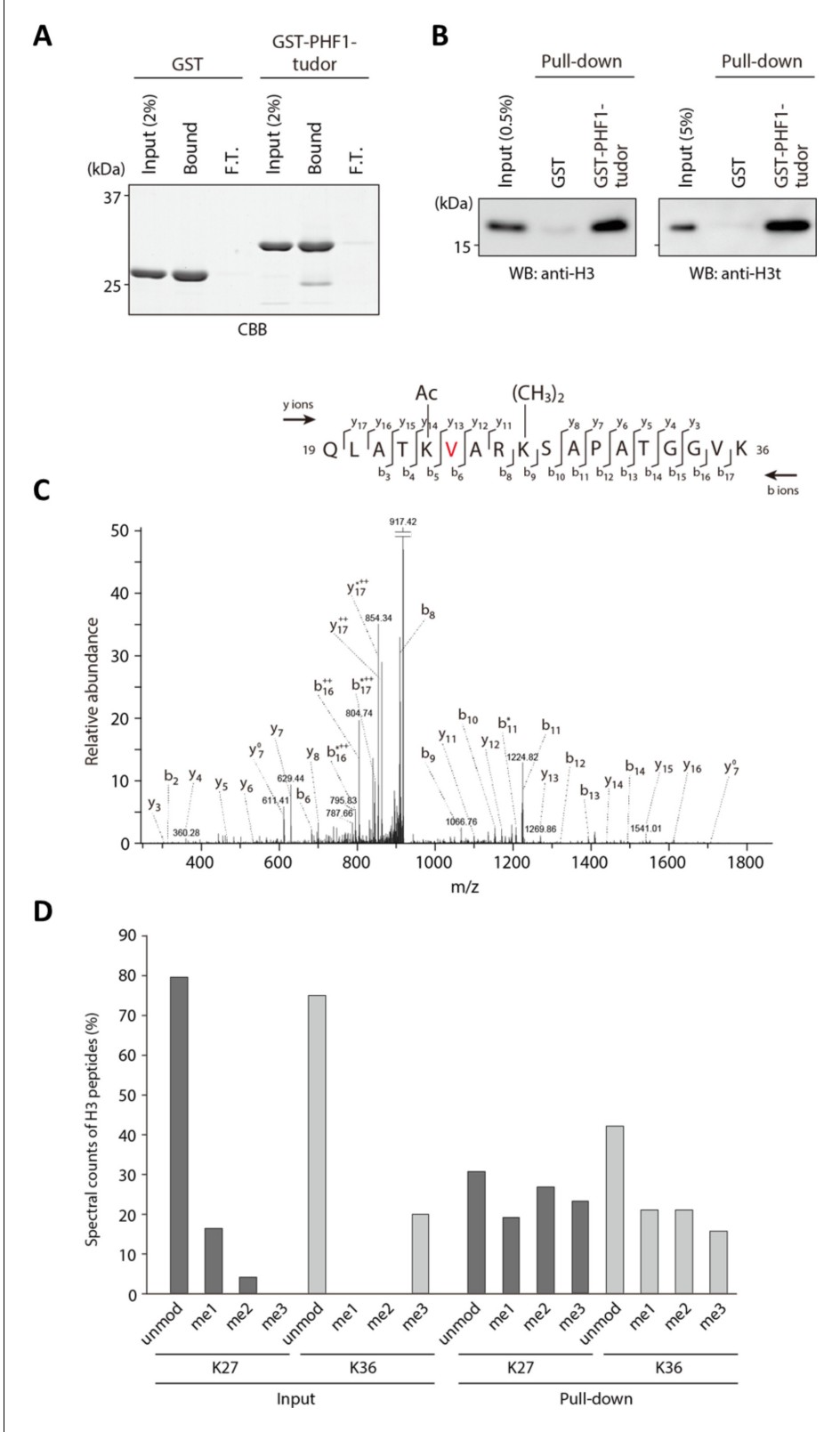

**Figure 3.** PHF1-Tudor could enrich H3t peptides. (**A, B**) In vitro GST pull-down assays using GST-tagged PHF1-Tudor proteins and histones prepared from H3t-expressed HEK293T cells. GST proteins from input, bound, and unbound flow-through (FT) fractions were analyzed using SDS-PAGE with Coomassie Brilliant Blue (CBB) staining (**A**). Input and bound histones were analyzed using western blotting with indicated antibodies (**B**). (**C**) MS/MS spectrum of the histone H3t peptide corresponding to residues 19–36. The observed *y* and *b* ions and fragment map are shown. (**D**) Spectral counts of

*Figure 3 continued on next page*

*Figure 3 continued*

methylated lysine residues in input or pull-down fractions. Control histones (Input) or pooled histones pulled-down with the GST-PHF1-Tudor (pull-down) were digested with LysN proteinase and subjected to LC-MS/MS analysis (detected peptides were summarized in *Supplementary file 1*). The spectral counts of methylated lysine residues in each fraction are shown. Since short peptides cleaved at unmodified lysine residues could not be efficiently detected in the LC-MS/MS analysis, these data do not accurately reflect their abundance.

## Structure-based engineering of the H3K36me3 peptide for enhanced PHF1 and PHF19 binding

Structural comparison of these PHF1 and PHF19 complexes reveals that, in addition to the aromatic cage recognition of the trimethylated lysine, the leucyl patch of PHF1/19 also makes a significant contribution to the histone H3 binding by its interactions with V24 of H3tK27me3 or H39 of H3K36me3. Considering that histidine is a slightly positively charged amino acid at physiological pH, we assumed that a histidine to valine substitution in H3K36me3 would enhance its binding to PHF1/19. Indeed, the H3K36me3-H39V mutant peptide exhibited approximately twofold enhanced binding to PHF1/19. On the other hand, the V24H mutation in H3tK27me3 led to a ~ tenfold loss in binding affinity to PHF1/19 (*Figure 6A*). The H3K9me3 peptide with a threonine residue at this position, or the canonical H3K27me3 peptide with an alanine at this position, has also been shown to have significantly weaker binding to PHF1/19 (*Figure 1D–E*; *Musselman et al., 2012*). Interestingly, when we replaced the R17 residue of the H4K20me3 peptide with valine, the mutation peptide also displayed stronger binding to PHF1/19 than both the wild-type and R17H H4K20me3 peptides

**Table 2.** Data collection and refinement statistics.

|  | PHF1-H3tK27me3 | PHF19-H3tK27me3 | PHF1-H3K36me3-H39V |
|---|---|---|---|
| PDB ID | 6WAU | 6WAT | 6WAV |
| Data collection |  |  |  |
| Space group | P2$_1$ | P3$_2$ | C2 |
| Cell (a,b,c (Å)) | 32.8, 286.8, 123.5 | 111.5, 111.5, 34.4 | 153.0, 65.6, 29.0 |
| (α, β, γ (°)) | 90, 90.7, 90 | 90, 90, 120 | 90, 97.1, 90 |
| Resolution (Å) | 47.80–1.80 | 55.74–1.71 | 40.07–1.70 |
|  | (1.83–1.80) | (1.74–1.71) | (1.73–1.70) |
| Rmerge overall | 0.071 (0.939) | 0.048 (1.314) | 0.048 (0.416) |
| Rmeas overall | 0.082 (1.117) | 0.054 (1.608) | 0.056 (0.487) |
| No. reflections | 207862 (9809) | 51838 (2547) | 29381 (1501) |
| Mean I/sigma | 10.5 (1.3) | 21.6 (0.7) | 16.4 (3.1) |
| CChalf | 0.999 (0.706) | 1.000 (0.190) | 0.998 (0.810) |
| Completeness (%) | 99.2 (93.7) | 99.4 (92.1) | 93.8 (90.9) |
| Multiplicity | 3.8 (3.3) | 5.4 (2.9) | 3.7 (3.6) |
| Model refinement |  |  |  |
| Resolution (Å) | 40.76–1.80 | 48.27–1.75 | 32.80–1.70 |
| Reflections/free | 395473/19949 | 48195/2242 | 29359/1455 |
| Rwork/Rfree | 0.215/0.260 | 0.209/0.251 | 0.198/0.242 |
| All (No. atoms/mean B (Å$^2$)) | 33004/39.0 | 3137/30.0 | 2348/20.8 |
| Tudor | 27847/38.7 | 2697/29.6 | 1878/19.5 |
| Peptide | 4169/42.0 | 383/32.2 | 306/26.0 |
| Water | 953/36.5 | 47/30.7 | 132/25.1 |
| Others | 35/31.9 | 10/26.7 | 32/27.4 |
| rmsd bonds (Å)/angles (°) | 0.011/1.1 | 0.012/1.9 | 0.011/1.7 |

Values in parentheses are for the highest-resolution shell.

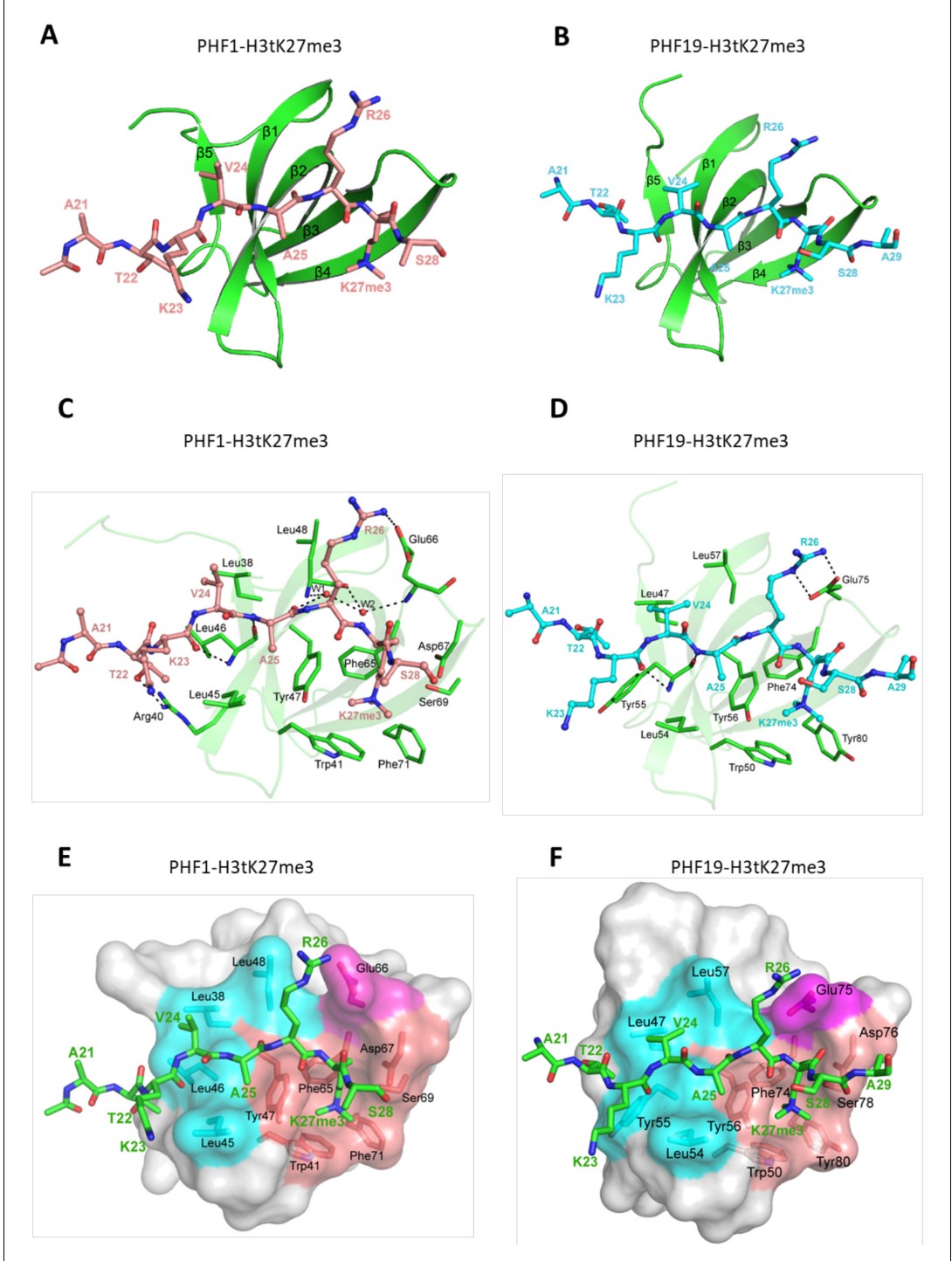

**Figure 4.** Crystal structures of PHF1/19-Tudor domain in complex with H3tK27me3. (A) Overall structure of PHF1-Tudor in complex with H3tK27me3. PHF1-Tudor is shown in cartoon representation and colored in green. The H3tK27me3 peptide is colored in salmon and shown as sticks. (B) Overall structure of PHF19-Tudor in complex with H3tK27me3. PHF19-Tudor is shown in cartoon representation and colored in green. The H3tK27me3 peptide is colored in cyan and shown as sticks. (C) Close-up view of the detailed interactions between PHF1-Tudor and the H3tK27me3 peptide. The H3tK27me3 peptide is shown as salmon sticks and the peptide interacting residues in PHF1 are shown as green sticks. The hydrogen bonds are indicated as dash lines and two involved water molecules are shown as spheres. (D) Close-up view of the detailed interactions between PHF19-Tudor and the H3tK27me3 peptide. The H3tK27me3 peptide and the peptide interacting residues in PHF19 are shown as cyan and green sticks, respectively. The hydrogen bonds are indicated as dash lines. (E-F) PHF1-Tudor and PHF19-Tudor are displayed in surface representations with the H3tK27me3

*Figure 4 continued on next page*

*Figure 4 continued*
peptide shown as green sticks. The three interaction regions of PHF1 and PHF19: hydrophobic patch, negative groove, and aromatic cage are colored in cyan, magenta, and salmon, respectively. The residues of PHF1/19 involved in the interactions are shown as sticks.

(*Figure 6A*). Taken together, valine is a favored residue type at this position, which could snugly complement the PHF1/19 leucyl patch.

To gain further insight into the leucyl patch interaction, we also determined the crystal structure of PHF1 in complex with the H3K36me3-H39V peptide (*Figure 6B*). As expected, the trimethylated lysine 36 is inserted into the same aromatic cage. V39 instead of H39 is located on the hydrophobic patch mimicking H3tK27me3-V24 (*Figure 6C*). P38 is structurally conserved in both complexes. In summary, the valine residue is an optimal residue in the recognition of the PHF1/19 Tudor domain. Among the five H3 variants, V24 is unique to H3t, and may be a key epigenetic element that distinguishes the testis histone H3 from other H3 variants, and the unique valine recognition by the leucyl patch of PHF1/19 could be used as a venue for the design of chemical probes and pharmaceutical drugs in the future.

## Discussion

The Tudor domain is a versatile histone binder, recognizing different histone PTMs, including H3K4me3, H3K9me3, H3K36me3, and H4K20me3 (*Lu and Wang, 2013*). The Tudor domain-containing PCL proteins are essential components of the PRC2 complex that maintains gene repression (*Margueron and Reinberg, 2011*), and the Tudor domains of PHF1/19 have been identified as readers of H3K36me3 in vitro (*Ballaré et al., 2012*; *Brien et al., 2012*; *Musselman et al., 2012*; *Qin et al., 2013*). In this study, we provide structural insights that the Tudor domains of PHF1/19 are also able to recognize testis-specific histone variant H3tK27me3 via a conserved aromatic cage, a salt-bridge, and a hydrophobic surface, and both H3tK27me3 and H3K36me3 engage the same binding surface of PHF1/19 but are bound in opposite orientations.

It has been well established that PRC2 plays an essential role in the regulation of mammalian spermatogenesis, including the maintenance of spermatogonial stem cells and the progression of meiosis (*Mu et al., 2014*). During spermatogenesis, undifferentiated spermatogonia first proliferate by mitosis to produce differentiating spermatogonia and eventually become spermatocyte to undergo meiosis. During spermatogonial differentiation, canonical histones are replaced by testis-specific histone variants and decorated with multiple post-translational modifications (*Govin et al., 2004*), including H3K4me3, H3K9me3, H3K27me3, and acetylation of H4 (*Boulay et al., 2011*; *Brykczynska et al., 2010*; *Rathke et al., 2007*). Of note, during spermatogenesis, BRDT (a testis-specific member), PHF1, EZH2, and EED are all highly expressed at the pachytene stage and early stage of the round spermatid (*Ha et al., 1991*; *Kawakami et al., 1998*; *Lambrot et al., 2012*; *Mu et al., 2014*; *Shang et al., 2007*). BRDT is responsible for recognizing H4ac (*Pivot-Pajot et al., 2003*), and EZH2 and EED catalyze H3K27 trimethylation. PHF1/19 is likely to read K27me3 of testis-specific variant H3t and spread the H3tK27me3 mark, leading to gene repression.

Previous studies have suggested that PHF1 and PHF19 co-localize with the K27me3 mark, not K36me3 in HEK293 cells (*Brien et al., 2012*; *Kycia et al., 2014*). Here, we provide further evidence that PHF1 co-localizes with H3tK27 in testis and has a preference toward H3tK27me3 over canonical H3K27me3 in vitro. Given that H3t plays an important role in the chromatin reprogramming and spermatogenesis (*Tachiwana et al., 2010*; *Tachiwana et al., 2008*; *Ueda et al., 2017*), PHF1 might also contribute to the mammalian spermatogenesis. As a matter of fact, in the male germ cells (GC1spg), knockdown of mouse PHF1 indeed results in a broad effect on the *Hox* gene silencing including *HoxA10* and a global decrease in the H3K27me3 level (*Cao et al., 2008*). Therefore, PHF1 might play a crucial role in gene silencing in the male germ cells.

The testis-specific histone variant H3t (also known as H3.4) has been identified in human, mouse, and rat genomes (*Maehara et al., 2015*). The human H3t contains only four amino acids (Val24, Met71, Ser98, and Val111) that are distinct from canonical histone H3.1 (Ala24, Val71, Ala98, and Ala111; *Figure 6D*). Two of them (Val24 and Ser98) are conserved among these species, of which Ser98 is located on the alpha helix of the histone-fold core in addition to Met71 and Val111 (*Tachiwana et al., 2010*). Val24 is in the N-terminal tail that is subject to extensive post-translational

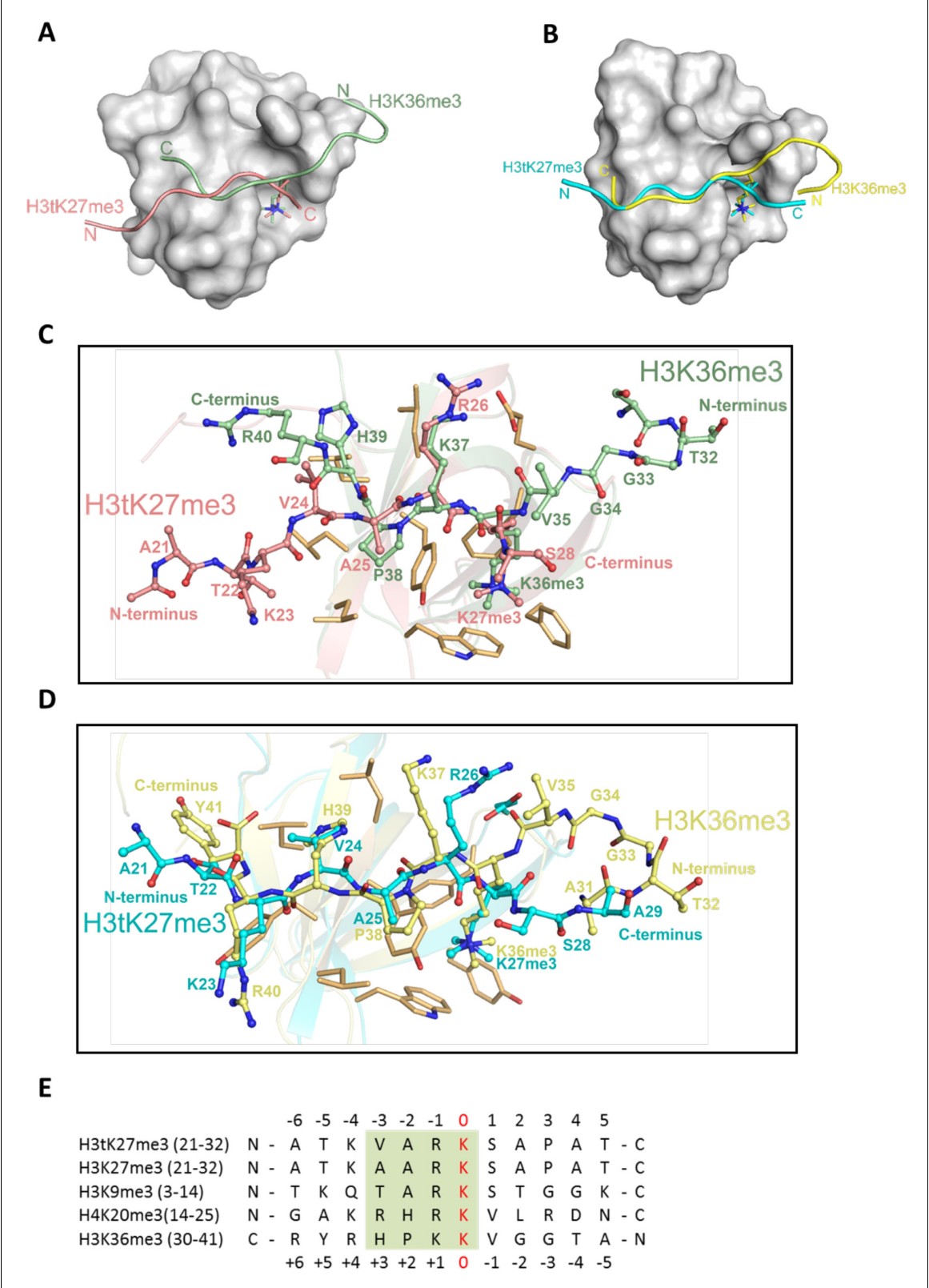

**Figure 5.** H3tK27me3 and H3K36me3 peptides bound to PHF1/19 within the same binding grooves but in reverse directions. (A) Superposition of the complex structures of PHF1-H3tK27me3 and PHF1-H3K36me3 (PDB: 4HCZ). The PHF1-Tudor is shown in a gray surface representation. The H3tK27me3 and H3K36me3 peptides are shown as salmon and pale green cartoons, respectively. (B) Superposition of the complex structures of PHF19-H3tK27me3 and PHF19-H3K36me3 (PDB: 4BD3). The PHF19-Tudor is shown in a gray surface representation. The H3tK27me3 and H3K36me3 peptides are shown as

*Figure 5 continued on next page*

*Figure 5 continued*

cyan and yellow cartoons, respectively. (C) Comparison of the detailed interactions of the H3tK27me3 and H3K36me3 peptides with PHF1. The H3tK27me3 and H3K36me3 peptides are shown in salmon and pale green sticks, respectively, and the key interacting residues of PHF1 are shown as light orange sticks. (D) Comparison of the detailed interactions of the H3tK27me3 and H3K36me3 peptides with PHF19. The H3tK27me3 and H3K36me3 peptides are shown in cyan and yellow sticks, respectively, and the key interacting residues of PHF19 are shown as light orange sticks. (E) Multiple sequence alignment of different peptides. The trimethylated lysines in the peptides are defined as position zero. The H3K36me3 sequence is aligned in a reverse direction. The relatively conserved motif is indicated in a green background.

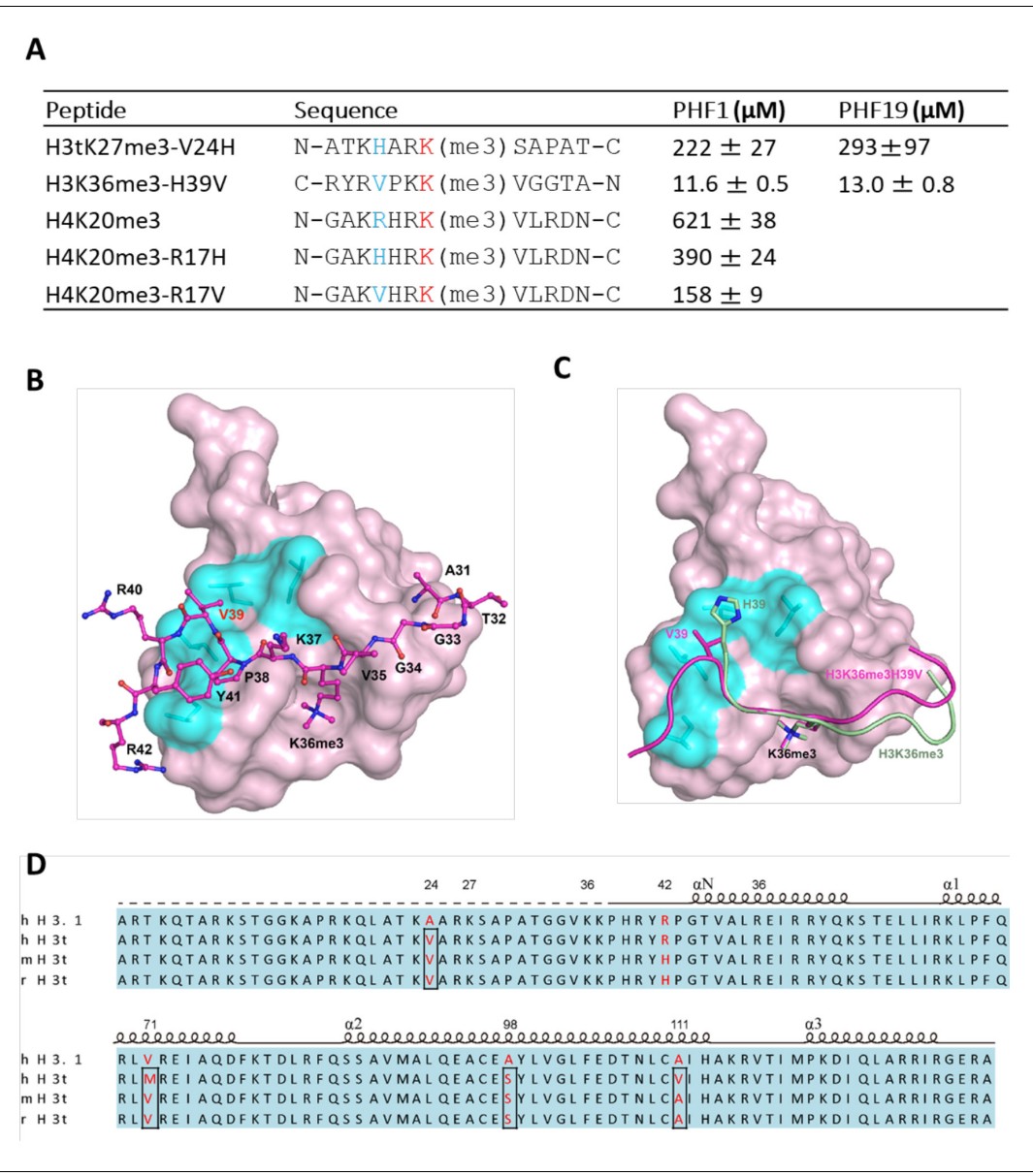

**Figure 6.** V24 of H3t is an optimal residue for the H3t recognition by the Tudor domain of PHF1/19. (A) Binding affinities of PHF1 and PHF19 Tudor domain with various peptides determined using ITC. (B) The crystal structure of PHF1 in complex with H3K36me3H39V. The mutant H3K36me3H39V peptide is shown as a stick in light magenta. PHF1 is shown as surface and the hydrophobic patch surrounding V39 is indicated in cyan. Some additional interactions in the H3K36me3-H39V mutant are indicated with dash lines. (C) Structural comparison of PHF1-H3K36me3 (PDB: 4HCZ) and PHF1-H3K36me3-H39V. The four leucines involved in hydrophobic interactions are shown with stick in cyan. The H3K36me3 peptide and H3K36me3-H39V mutant are shown as cartoon in pale green and light magenta, respectively. (D) The sequence alignment of H3t from human, mouse, and rat. The secondary structure elements are shown above the sequence. Identical residues are indicated in aqua and the different residues are colored in red.

modifications. Therefore, the conserved Val24 may play a critical role in chromatin signaling by distinguishing testis-specific histone H3t from other H3 variants. In this study, we identified the Tudor domains of PHF1/19 could specifically recognize the H3tK27me3 mark and Val24 plays a pivotal role in its recognition by the PHF1/19 Tudor domains through a leucyl patch. H3t is initially just found in the testis (*Witt et al., 1996*) but later studies revealed that H3t is also present in other cells (*Andersen et al., 2005*; *Govin et al., 2005*; *Kycia et al., 2014*). For instance, H3t is also abundantly expressed in the brain and embryo (*Govin et al., 2007*; *Tachiwana et al., 2008*). Hence, PHF1/19 may have a wide range of functions in mammals, which remains to be investigated.

# Materials and methods

## Key resources table

| Reagent type (species) or resource | Designation | Source or reference | Identifiers | Additional information |
|---|---|---|---|---|
| Gene (*Mus musculus*) | H3.4 histone (H3t) | GenBank | 382523 | |
| Gene (*Homo sapiens*) | PHF1 | GenBank | 5252 | |
| Strain, strain background (*Escherichia coli*) | BL21(DE3) | Thermo Fisher | C600003 | |
| Cell line (*Homo sapiens*) | 293T/17 | ATCC | CRL-11268 | |
| Antibody | Anti-histone H3 (rabbit polyclonal) | abcam | ab1791 | WB (1:2000) |
| Antibody | Anti-histone H3t (mouse monoclonal) | This paper *Ueda et al., 2017* | | WB (1:1000) IF (1:1000) |
| Antibody | Anti-PHF1 (rabbit polyclonal) | Abcam | Cat# ab80042 | IF (1:500) |
| Antibody | Anti-SCP3 (mouse monoclonal) | Abcam | 10G11/7 Cat# ab97672 | IF (1:1000) |
| Antibody | Anti-γ H2A.X (mouse monoclonal) | Merck Millipore | Cat# #05–636 | IF (1:1000) |
| Antibody | Lectin PNA tagged with Alexa Fluor 488 | Thermo Fisher | Cat# L21409 | IF (1:1000) |
| Sequence-based reagent | Eef1a1–Fw | *Ueda et al., 2017* | PCR primers | CTCTGACTACCCTCCACTTGGTCG |
| Sequence-based reagent | Eef1a1–Rev | *Ueda et al., 2017* | PCR primers | ATTAAGACTGGGGTGGCAGGTGTT |
| Sequence-based reagent | Phf1–Fw | This paper | PCR primers | TGAGAAGTGTCGCCATGCTTA |
| Sequence-based reagent | Phf1–Rev | This paper | PCR primers | CATAGGGACCTTTCTTCAGTGC |
| Recombinant DNA reagent | pET28-MHL | SGC, Toronto | 26096 (Genbank accession number: EF456735) | Expression of the Tudor domains of PHF1, MTF2 and PHF19, and their mutants |
| Software, algorithm | Origin 6.1, Origin 7.0 | OriginLab | http://www.originlab.com/ | For ITC curve fitting and calculation of Kd values |
| Software, algorithm | XDS | PMID:20124692 | http://xds.mpimf-heidelberg.mpg.de/ | Processing of X-ray diffraction data |
| Software, algorithm | Pymol | DeLano Scientific LLC | http://www.pymol.org/ | Drawing structure figures |
| Software, algorithm | Coot | PMID:15572765 | http://www2.mrc-lmb.cam.ac.uk/Personal/pemsley/coot/ | Structure model building |

*Continued on next page*

*Continued*

| Reagent type (species) or resource | Designation | Source or reference | Identifiers | Additional information |
|---|---|---|---|---|
| Software, algorithm | CCP4 | PMID:21460441 | https://www.ccp4.ac.uk/ | Structure determination and refinement |
| Software, algorithm | MASCOT program | Matrix Science | | Version 2.6 |
| Software, algorithm | Proteome discoverer 2.2 | Thermo Fisher | | |
| Other | Hoechst 33342 | Thermo Fisher | Cat# H3570 | IF (1:5000) |

## Protein expression and purification

The genes coding for PHF1-Tudor (aa 28–87), MTF2-Tudor-PHD1 (aa 41–158), and PHF19-Tudor (aa 38–96) were ligated into a pET28a-MHL expression vector, respectively. The recombinant plasmids were transformed into *E. coli* BL21 (DE3) competent cells and overexpressed by 0.2 mM IPTG at 18° C overnight. The cells were harvested and lysed by sonication. The protein was purified by Ni-NTA and the N-terminal His tags were cleaved by TEV protease at 4°C overnight. The tags and protease were removed by reverse Ni-NTA affinity, then the flow through was applied to the anion exchange column and further purified by gel filtration chromatography. The gel filtration buffer contains 20 mM Tris-HCl pH7.5, 150 mM NaCl, and 0.5 mM TCEP. The mutant proteins were purified as the same protocol. Recombinant GST-fused-proteins used in the pull-down assays were expressed in BL21 (DE3) *E. coli* and purified by Glutathione Sepharose 4B (GE Healthcare), according to the manufacturer's instructions.

## Crystallization and structure determination

The proteins were incubated with peptides for 2 hr on ice before setting up the crystallization trials. The sitting drop vapor diffusion was used for screening the crystals by mixing 1 μL protein and 1 μL reservoir solution. The crystals of PHF1 in complex H3tK27me3 were grown in 2.4 M ammonium phosphate dibasic and 0.1 M HEPES pH 7.0 at 18°C. The crystals of PHF1 in complex with H3K36me3H39V were obtained in 1.2 M ammonium sulfate and 0.1 M sodium acetate 4.6 at 4°C. The crystals of PHF19 in complex H3tK27me3 were grown in 2.3 M ammonium phosphate dibasic and 0.1 M Tris pH 8.5 at 18°C. The crystallization condition supplemented with 20% (v/v) glycerol was used as cryoprotectant. The crystals were protected in cryoprotectant and flash-frozen in liquid nitrogen before data collection.

PHF1-H3tK27me3: Diffraction data were collected at a copper rotating anode source and on beamline 08id of the Canadian Light Source. Diffraction images suggested the presence of multiple crystal lattices in the sample. Diffraction images were reduced to intensities with XDS (*Kabsch, 2010*) and symmetry-related intensities merged with AIMLESS (*Evans and Murshudov, 2013*). We selected a primitive, pseudo-orthorhombic indexing solution with an unusually long unique axis. Beginning with coordinates from PDB entry 4HCZ, we used the programs PHASER (*McCoy et al., 2007*) and MOLREP (*Vagin and Teplyakov, 2010*), as well as visual inspection of difference electron density, to locate 30 copies of the Tudor domain in the asymmetric unit. PHF1 monomers assemble into homodimers that are replicated through translational pseudosymmetry. We used REFMAC (*Murshudov et al., 2011*) and PHENIX (*Liebschner et al., 2019*) for restrained refinement of model coordinates and thermal displacement parameters and COOT for interactive rebuilding (*Emsley et al., 2010*).

PHF19-H3tK27me3: Diffraction data were collected at beamline 08b1 of the Canadian Light Source and reduced with XDS/AIMLESS. In the course of structure determination, we (subsequently) used data sets collected from two crystals. The structure was solved by molecular replacement with the program PHASER and a mutated search model based on coordinates from PDB entry 4HCZ. Molecular replacement search and subsequent refinement revealed problems with the placement of an expected third copy of the Tudor domain in the P62 unit. Inspection of difference electron density after the expansion of the asymmetric unit to space group P32 suggested alternate positions of the 'third' (symmetry P62) or fifth and sixth (P32) copies of the Tudor domain. We used twinned refinement modes in REMAC and PHENIX with twin law [-h, -k,l] but also inspected maps calculated

in AUTOBUSTER (*Smart et al., 2012*), for which we did not explicitly select twin refinement mode. We used COOT for manual model (re-)building.

PHF1-H3K36me3-H39V: Diffraction data were collected on a copper rotating anode and reduced with XDS/AIMLESS. Data reduction was controlled through the XIA2 (*Winter et al., 2013*) interface for some runs. In the course of structure determination, we (subsequently) used data sets collected from two crystals. The structure was solved by molecular replacement with the program PHASER and coordinates from PDB entry 4HCZ. The model was refined with REFMAC and interactively rebuilt with COOT.

## Isothermal titration calorimetry

Isothermal titration calorimetry (ITC) measurements were carried out at 25˚C using a MicroCal VP-ITC instrument. The ITC buffer contains 20 mM Tris-HCl pH 7.5, 150 mM NaCl, and 0.5 mM TECP. The final concentration of protein and peptide was 60–100 µM and 1–5 mM, respectively. The peptide was titrated into the protein solution with 26–28 injections of 10 µL each. Injections were spaced 180 s with a reference power of 15 µcal/s. The ITC data were processed using Origin software.

## Cell culture

HEK293T cells were cultured in DMEM (Nacalai Tesque) supplemented with 10% fetal calf serum (Invitrogen). The HEK293T cells were purchased from ATCC, have been tested negative for mycoplasma contamination and aliquots were made for stocks.

## Plasmids

To produce mouse H3t in HEK293T cells, cDNA was PCR amplified using KOD-Plus-Neo (Toyobo). The PCR products were cloned into the pCRII vector using the TOPO-TA cloning kit (Invitrogen), sequenced, and subcloned into pcDNA4/TO/puro, a pcDNA4/TO derivative containing the puromycin resistant gene. To produce PHF1-Tudor as GST fusion proteins in *E. coli*, the portion of the PHF1 coding sequence corresponding to amino acids 28–87 was PCR amplified, cloned, and sequenced as above, and subcloned into the pGEX-6P-3 (GE Healthcare).

## Antibodies

The following antibodies were used in this study: anti-histone H3 (rabbit polyclonal, ab1791; Abcam, Cambridge, MA), anti-histone H3t (mouse monoclonal; *Ueda et al., 2017*), anti-PHF1 (rabbit polyclonal, ab80042; Abcam; *Qu et al., 2018*), anti-SCP3 (mouse monoclonal, 10G11/7, ab97672; Abcam), anti-γ H2A.X (mouse monoclonal, #05–636; Merck Millipore) antibodies were used. For staining of sperm acrosomes, lectin PNA tagged with Alexa Fluor 488 (Thermo Fisher Scientific Inc) was used.

## Animals

Mature male C57BL6/N mice (over 10 weeks of age) were purchased from Japan SLC Inc (Hamamatsu, Japan). All animal care was in accordance with the institutional guidelines and was approved by the Institutional Animal Care and Use Committee of Asahikawa Medical University (No. 19102, 19156).

## Reverse transcription and real-time PCR quantification of mRNA

Total RNA was isolated from tissues of three male mice using the TRIzol Reagent (Thermo Fisher Scientific Inc) and cDNA was synthesized using PrimeScript RT Master Mix (Takara Bio Inc, Tokyo, Japan) as suggested by the manufacturers. Primer sequences for real-time PCR analyses were designed using the PrimerBank (https://pga.mgh.harvard.edu/primerbank/). Real-time PCR was performed using the SYBR Green PCR master mix (Takara Bio Inc) according to the manufacturer's specifications and fluorescent dye incorporation was used to calculate the critical threshold cycle number. Critical threshold values were normalized to the values for Eef1a1 and data are presented as the means plus the standard deviations. The sequences of the primers used were as follows: Eef1a1–Fw, 5′–CTCTGACTACCCTCCACTTGGTCG–3′; Eef1a1–Rev, 5′–ATTAAGACTGGGG TGGCAGGTGTT–3′; Phf1–Fw, 5′–TGAGAAGTGTCGCCATGCTTA–3′; Phf1–Rev, 5′–CATAGGGACC TTTCTTCAGTGC–3′.

## Histological analyses

For the staining of testicular sections with antibodies, the samples were first fixed with 4% paraformaldehyde at 4°C overnight, equilibrated in phosphate-buffered saline (PBS) (–) with 30% sucrose, embedded into OCT compound (Sakura Finetek, Tokyo, Japan) and frozen, then sectioned at 7 µm. Before staining, fixed sections were first activated with boiled citrate buffer (pH 6.0). All sections were stained with Hoechst 33342 (Thermo Fisher Scientific Inc) and were imaged using a BZ-X700 microscope (Keyence Corp., Osaka, Japan) or FV1000D confocal laser scanning microscope (Olympus Corp., Tokyo, Japan).

## Preparation of histones from HEK293T cells

The histone extraction was performed as previously described (*Sadaie et al., 2008*) with some modifications. H3t-expressing plasmid (pcDNA4/TO/puro-H3t) was introduced into HEK293T cells using Lipofectamine 3000 reagent (Invitrogen), and after 72 hr, the cells (semi-confluent, 10 × 100 mm dishes) were washed twice with phosphate-buffered saline and then lysed in 1 mL (per dish) of nuclear lysis buffer (100 mM Tris-HCl pH 7.5, 150 mM NaCl, 1.5 mM MgCl$_2$, 0.65% Nonidet P-40, and 1 mM phenylmethylsulfonyl fluoride) supplemented with proteinase inhibitor mixture (cOmplete EDTA-free; Roche). The nuclei were harvested by scraping into a 1.5 mL microcentrifuge tube and collected by centrifugation at 500 × g for 5 min at 4°C. After removal of the supernatant, the nuclei were resuspended in 200 µL of 0.4 N H$_2$SO$_4$ and the suspension was further incubated with rotation for 2 hr at 4°C. The insoluble fraction was removed by centrifugation at 5500 × g for 5 min at 4°C. The acid-soluble supernatant was collected in a new centrifuge tube and the proteins were precipitated with 50 µL of 100% trichloroacetic acid (final concentration, 20%). The suspension was placed on ice for 1 hr and centrifuged at 22,000 × g for 15 min at 4°C. The protein pellet was washed three times with acetone, air-dried for 5 min, and resuspended in 50–100 µL of deionized H$_2$O by rotating at 4°C overnight. After the recovery was estimated, the extracted proteins were stored at −80°C until use.

## GST pull-down assays

GST or GST-PHF1-Tudor proteins (10 µg each) were incubated with 4 µL (25% slurry) of Glutathione Magnetic Beads (88821; Thermo Scientific) in 250 µL of binding buffer (125 mM Tris-HCl pH8.0, 150 mM NaCl, and 0.05% Tween 20) with rotating for 2 hr at 4°C. The GST- or GST-PHF1-Tudor-bound beads were washed three times with binding buffer and once with reaction buffer (10 mM Tris-HCl pH7.6, 150 mM NaCl, 0.1 mM EDTA, and 0.1% Triton X-100) supplemented with proteinase inhibitor mixture (cOmplete; Roche), and incubated with 5 µg of histone in 250 µL of reaction buffer at 4°C for 2 hr. After the beads were washed three times with the reaction buffer, bound proteins were eluted by boiling with 1 × SDS sample buffer. The eluted proteins were resolved by 15% SDS-PAGE and the pulled-down histones were detected by western blotting. For LC-MS/MS analysis, the GST pull-down assay using GST-PHF1-Tudor was repeated 10 times, pooled pulled-down histones were combined, resolved by 15% SDS-PAGE, and subjected to LC-MS/MS analysis.

## Nano-liquid chromatography-tandem mass spectrometry (LC-MS/MS)

The proteins in each gel slice were subjected to reduction with 10 mM dithiothreitol (DTT), at 56°C for 1 hr, alkylation with 55 mM iodoacetamide at room temperature for 45 min in the dark, and digestion with 250 ng of LysC (Wako) or 250 ng of LysN (Promega) at 37°C for 16 hr. The resulting peptides were extracted with 1% trifluoroacetic acid and 50% acetonitrile, dried under a vacuum, and dissolved in 2% acetonitrile and 0.1% formic acid. The peptides were then fractionated by C18 reverse-phase chromatography (ADVANCE UHPLC; AMR Inc) and applied directly into a hybrid linear ion trap mass spectrometer (LTQ Orbitrap Velos Pro; Thermo Fisher Scientific) with Advanced Captive Spray SOURCE (AMR Inc). The mass spectrometer was programmed to carry out 11 successive scans, with the first consisting of a full MS scan from 350 to 1600 m/z using Orbitrap at a resolution of 60,000 and the second to eleventh consisting of data-dependent scans of the top ten abundant ions obtained in the first scan using ion trap. Automatic MS/MS spectra were obtained from the highest peak in each scan by setting the relative collision energy to 35% and the exclusion time to 90 s for molecules in the same m/z value range. The molecular masses of the resulting peptides were searched against the Uniprot Human proteome database (2018.12.13 downloaded)

including the Mouse H3t sequence (PDB id: 5B1L-A) and cRAP contaminant proteins dataset using the MASCOT program (version 2.6; Matrix Science) via Proteome discoverer 2.2 (Thermo Fisher Scientific). The search was performed using carbamidomethylation of cysteine as a fixed modification, and oxidation of methionine, acetylation of protein N-termini, acetylation, methylation, demethylation, and trimethylation of lysine as variable modifications. The number of missed cleavages sites was set as 4.

## Acknowledgements

We thank Wolfram Tempel for assistance in data collection and structure determination, David Waterman and Andrey Lebedev for comments on the PHF1-H3tK27me3 diffraction data, Machika Kawamura and Chen Chen for technical help, and Yasuyuki Ohkawa for anti-H3t antibody. The SGC is a registered charity (number 1097737) that receives funds from AbbVie, Bayer Pharma AG, Boehringer Ingelheim, Canada Foundation for Innovation, Eshelman Institute for Innovation, Genome Canada through Ontario Genomics Institute [OGI-055], Innovative Medicines Initiative (EU/EFPIA) [ULTRA-DD grant number 115766], Janssen, Merck KGaA, Darmstadt, Germany, MSD, Novartis Pharma AG, Ontario Ministry of Research, Innovation and Science (MRIS), Pfizer, São Paulo Research Foundation-FAPESP, Takeda, and Wellcome. This work was also supported by the National Natural Science Foundation of China grant 31900865 (CD), KAKENHI [18H05532 (JN), 19K06452 (JU)], the Takeda Science Foundation (JU), Kato Memorial Bioscience Foundation (JU), and Akiyama Life Science Foundation (JU). This work was also supported by the State Key Laboratory of Medicinal Chemical Biology (NanKai University). Part of the research described in this paper was performed at the Canadian Light Source, a national research facility of the University of Saskatchewan, which is supported by the Canada Foundation for Innovation (CFI), the Natural Sciences and Engineering Research Council (NSERC), the National Research Council (NRC), the Canadian Institutes of Health Research (CIHR), the Government of Saskatchewan, and the University of Saskatchewan.

## Additional information

### Funding

| Funder | Grant reference number | Author |
|---|---|---|
| National Natural Science Foundation of China | 31900865 | Cheng Dong |
| Japan Society for the Promotion of Science KAKENHI grants | 18H05532 | Jun-ichi Nakayama |
| Japan Society for the Promotion of Science KAKENHI grants | 19K06452 | Jun Ueda |
| Takeda Science Foundation | | Jun Ueda |
| Kato Memorial Bioscience Foundation | | Jun Ueda |
| Akiyama Life Science Foundation | | Jun Ueda |

The funders had no role in study design, data collection and interpretation, or the decision to submit the work for publication.

### Author contributions

Cheng Dong, Conceptualization, Data curation, Formal analysis, Funding acquisition, Validation, Investigation, Writing - original draft, Writing - review and editing; Reiko Nakagawa, Kyohei Oyama, Yusuke Yamamoto, Yuriko Yoshimura, Hiroyuki Kamiya, Formal analysis, Investigation; Weilian Zhang, Yanjun Li, Investigation; Aiping Dong, Validation, Investigation; Jun-ichi Nakayama, Jun Ueda, Formal analysis, Supervision, Funding acquisition, Writing - original draft, Writing - review and

editing; Jinrong Min, Conceptualization, Formal analysis, Supervision, Funding acquisition, Writing - original draft, Project administration, Writing - review and editing

### Author ORCIDs
Cheng Dong  https://orcid.org/0000-0002-2891-8759
Reiko Nakagawa  http://orcid.org/0000-0002-6178-2945
Jun Ueda  https://orcid.org/0000-0002-9766-203X
Jinrong Min  https://orcid.org/0000-0001-5210-3130

### Decision letter and Author response
Decision letter https://doi.org/10.7554/eLife.58675.sa1
Author response https://doi.org/10.7554/eLife.58675.sa2

## Additional files

### Supplementary files
• Supplementary file 1. Summary of H3K27 residue containing H3/H3t peptides detected in the input fraction.

• Transparent reporting form

### Data availability
Diffraction data have been deposited in PDB under the accession codes 6WAT, 6WAU, 6WAV.

The following datasets were generated:

| Author(s) | Year | Dataset title | Dataset URL | Database and Identifier |
| --- | --- | --- | --- | --- |
| Dong C, Bountra C, Edwards AM, Arrowsmith CH, Min JR | 2020 | Complex structure of PHF1 | https://www.rcsb.org/structure/6WAT | RCSB Protein Data Bank, 6WAT |
| Dong C, Bountra C, Edwards AM, Arrowsmith CH, Min JR, Structural Genomics Consortium (SGC) | 2020 | Complex structure of PHF19 | https://www.rcsb.org/structure/6WAU | RCSB Protein Data Bank, 6WAU |
| Dong C, Bountra C, Edwards AM, Arrowsmith CH, Min JR, Structural Genomics Consortium (SGC) | 2020 | Crystal structure of PHF1 in complex with H3K36me3 substitution | https://www.rcsb.org/structure/6WAV | RCSB Protein Data Bank, 6WAV |

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
