## [Decision Letter]

**Acceptance summary:**

The Polycomb-like proteins PHF1, MTF2 and PHF19 are critical components of the (Polycomb repressive complex 2) complex PRC2 and stimulate the catalytic activity of PRC2 for gene silencing. Previously studies showed that the Tudor domains of PHF1/19 binds to H3K36me3 but not H3K27me3 in vitro. However, in cells the PHF1 and PHF19 proteins largely co-localize with the H3K27me3 mark, but not H3K36me3. The finding that the PHF1/19 Tudor domains preferentially bind to K27me3 on testis specific histone H3 (H3tK27me3) over canonical H3K27me3 reveals a novel role of the Polycomb-like proteins in testis. The complex structures of the PHF1/19 Tudor domains with H3tK27me3 shed light on the molecular basis for preferential recognition.

**Decision letter after peer review:**

Thank you for submitting your article "Structural basis for histone variant H3tK27me3 recognition by PHF1 and PHF19" for consideration by *eLife*. Your article has been reviewed by two peer reviewers, and the evaluation has been overseen by a Reviewing Editors and Cynthia Wolberger as the Senior Editor. The following individual involved in review of your submission has agreed to reveal their identity: Tatiana Kutateladze (Reviewer #2).

The reviewers have discussed the reviews with one another and the Reviewing Editor has drafted this decision to help you prepare a revised submission.

Summary:

In this manuscript, Dong and colleagues investigate binding selectivity of the Tudor domain from PHF1, PHF19 and MTF2, particularly to the methylated H3K27, H3K36 and H3tK27 sites. The authors determined the crystal structures of the Tudor domains of PHF1 and PHF19 in complex with H3tK27me3 peptide and show that these Tudor domains bind comparably to H3tK27me3 and H3K36me3. The authors also demonstrate that PHF1 co-localizes with H3t during pachytene and diplotene stages and the round spermatid stage. Overall, the manuscript describes novel findings that are of general interest to the chromatin field. Although the cell-based evidence is ambiguous and limited, the in vitro biochemical and structural data is with excellent quality that support the overall conclusions.

Essential revisions:

1) Based on the structures described here and the structures of the PRC2 complexes, it would be interesting to know the orientation of the Tudor domain with respect to H3K27me and H3K36me. To show a model would be exciting (optional).

2) Please describe what structures are overlaid in Figure 1G.

3) Please provide H3t staining in Figures 2C and 2D.

4) Figure 3D: Does the MS distinguish K27me from K36me? Please describe how many biological repeats this figure represents and include the raw data as supplementary data.

---

## [Author Response]

Essential revisions:1) Based on the structures described here and the structures of the PRC2 complexes, it would be interesting to know the orientation of the Tudor domain with respect to H3K27me and H3K36me. To show a model would be exciting (optional).

Thanks for the suggestion. To our knowledge, although the N-terminal (aa 38-95) and the C-terminal (aa 500-580) structures of PHF19 have been respectively determined, the full-length structure of PHF1/19 is still unknown. The C-terminal region is responsible for binding to SUZ12 in the PRC2 complex structure, and the N-terminal Tudor domain binds to H3K27me3 or H3K36me3, it is hard to put them together in a holo-PRC2 complex context in the absence of internal regions.

2) Please describe what structures are overlaid in Figure 1G.

Thanks for the advice. We have added the structure information in the figure legend.

3) Please provide H3t staining in Figures 2C and D.

Thanks for the advice. We have added the staining data in the revised manuscript Figure 2. As H3t is not a meiosis marker, we could not distinguish between leptotene and zygotene stage with high confidence. Thus, we did not include these two stages in the revised Figure 2.

4) Figure 3D: Does the MS distinguish K27me from K36me? Please describe how many biological repeats this figure represents and include the raw data as supplementary data.

The MS can distinguish K27me from K36me, we show one example for a peptide (Query 1204) containing both K27me1 and K36me3 in the following figure. As the detection efficiencies of modified peptides are generally low, we repeated the pull-down experiments as shown in the Figure 3B ten times, pooled the eluted fractions, and subjected to the MS/MS analysis. The obtained mass results are derived from the pooled samples. We have described it in the Materials and methods section. While we have used several peptidase (alone or combined) and tried to detect peptides containing A24/V24, K27, and K36, the number of such peptides were significantly low. So we use the results of Lys-N and summarized the results in Figure 3D. We have included the summary of identified peptides in Supplementary file 1.

**Author response image 1. sa2fig1:**